# The *Arabidopsis thaliana* mobilome and its impact at the species level

**Leandro Quadrana[1], Amanda Bortolini Silveira[1], George F Mayhew[2], Chantal LeBlanc[3], Robert A Martienssen[4,5], Jeffrey A Jeddeloh[2], Vincent Colot[1]***

[1]Institut de Biologie de l'Ecole Normale Supérieure, Centre National de la Recherche Scientifique, Institut National de la Santé et de la Recherche Médicale, Ecole Normale Supérieure, Paris, France; [2]Roche NimbleGen, Inc, Madison, United States; [3]Department of Molecular, Cellular and Developmental Biology, Yale University, New Haven, United States; [4]Watson School of Biological Sciences, Howard Hughes Medical Institute, Gordon and Betty Moore Foundation, Palo Alto, United States; [5]Cold Spring Harbor Laboratory, Cold Spring Harbor, United States

**Abstract** Transposable elements (TEs) are powerful motors of genome evolution yet a comprehensive assessment of recent transposition activity at the species level is lacking for most organisms. Here, using genome sequencing data for 211 *Arabidopsis thaliana* accessions taken from across the globe, we identify thousands of recent transposition events involving half of the 326 TE families annotated in this plant species. We further show that the composition and activity of the 'mobilome' vary extensively between accessions in relation to climate and genetic factors. Moreover, TEs insert equally throughout the genome and are rapidly purged by natural selection from gene-rich regions because they frequently affect genes, in multiple ways. Remarkably, loci controlling adaptive responses to the environment are the most frequent transposition targets observed. These findings demonstrate the pervasive, species-wide impact that a rich mobilome can have and the importance of transposition as a recurrent generator of large-effect alleles.

*For correspondence: colot@biologie.ens.fr

## Introduction

Transposable elements (TEs) are sequences that move and replicate around the genome. Depending on whether their mobilization relies on a RNA or DNA intermediate, they are classified as retrotransposons (class I) or DNA transposons (class II), respectively (*Slotkin and Martienssen, 2007*). TEs are further subdivided into distinct families, the prevalence of which differs between organisms because of a complex array of factors, including variable transposition activity and diverse selection pressures (*Barrón et al., 2014*). Given their mobile nature, TEs pose multiple threats to the physical and functional integrity of genomes. In particular, TEs can disrupt genes through insertion and also through excision in the case of DNA transposons. Thus, TE mobilization is a source of both germline and somatic mutations (*Richardson et al., 2015*; *Barrón et al., 2014*; *Lisch, 2013*). Although TEs are endogenous mutagens with potentially catastrophic effects, their mobilization might sometimes be beneficial. In fact, soon after their discovery, Barbara McClintock named TEs 'controlling elements' to emphasize their role in the control of gene action (*McClintock, 1956*). In mammals, transposition of some *LINE 1* retrotransposons occurs extensively during embryogenesis as well as in the adult brain, again suggesting functional relevance of somatic TE mobilization (*Richardson et al., 2014*). Nonetheless, TEs are under tight control to limit their mutational impact both within and across generations. In plants and mammals, a major control is through epigenetic silencing mechanisms, including DNA methylation and these mechanisms can in turn have 'epimutagenic' effects on adjacent

genes (*Slotkin and Martienssen, 2007*; *Weigel and Colot, 2012*; *Heard and Martienssen, 2014*; *Quadrana and Colot, 2016*).

Despite the many documented short-term as well as evolutionary consequences of TE mobilization (*Rebollo et al., 2012*; *Trono, 2016*; *Kazazian, 2004*; *Elbarbary et al., 2016*; *Bennetzen and Wang, 2014*; *Lisch, 2013*), TEs are among the least investigated components of genomes, mainly because they are present in multiple, often degenerated copies, which complicate analysis. Thus, a species-wide view of the mobilome - i.e. of the set of TE families with transposition activity - is lacking for most organisms.

Studies in humans suggest that although at least half of the 3Gb genome is made up of TE sequences, mainly belonging to *LINE 1* and *SINE* retrotransposon families, a few of these only and none of the other TE families contain mobile copies (*Richardson et al., 2015*). In contrast, the number of TE families that have retained transposition activity is much larger in the mouse and these include several so-called endogenous retroviruses (ERVs) in addition to *LINEs* and *SINEs* (*Richardson et al., 2015*). In Drosophila, which has a much smaller genome (~120 Mb) characterized by a large repertoire of class I and class II TE families, the situation is again different with most TE families likely mobile (*Mackay et al., 2012*; *Cridland et al., 2013*; *Robert et al., 2015*; *Rahman et al., 2015*). However, in this species and even more so in mammals, the population genetics of the mobilome remains poorly characterized.

The flowering plant *A. thaliana* is particularly attractive for conducting a systematic survey of the mobilome and of its molecular as well as phenotypic impact at the species level. First, like Drosophila, *A. thaliana* has a compact genome and a large repertoire of class I and class II TE families (*Arabidopsis Genome Initiative, 2000*; *Ahmed et al. 2011*; *Joly-Lopez and Bureau, 2014*). Thus, most TE families are of relatively small size, which facilitates their study. Second, *A. thaliana* occupies a wide range of habitats across the globe and representative accessions have been extensively characterized both genetically and phenotypically (*Weigel and Nordborg, 2015*). Third, whole genome sequencing has been performed for >1000 *A. thaliana* accessions and DNA methylome as well as transcriptome data are also available for hundreds of these (*Schmitz et al., 2013*; *Long et al., 2013*; *Dubin et al., 2015*; *Cao et al., 2011*). Finally, genome-wide association studies (GWASs) are straightforward in this species (*Weigel and Nordborg, 2015*).

Here we present a comprehensive assessment of the *A. thaliana* mobilome, which radically changes the prevailing view of limited transposition potential in this species and provides important novel insights into the population genetics of TE mobilization (*Hu et al., 2011*; *Maumus and Quesneville, 2014*). Specifically, we show that the *A. thaliana* mobilome is composed of a very large number of class I and class II TE families overall, but differs extensively among accessions. We further show that TE mobilization is a complex trait and we have identified environmental as well as genetic factors that influence transposition in nature. These factors include the annual temperature range, the TE themselves and multiple gene loci, notably *MET2a*, which encodes a poorly characterized DNA methyltransferase. In addition, we present compelling evidence that TEs insert throughout the genome with no overt bias and that the mobilome has a pervasive impact on the expression and DNA methylation status of adjacent genes. These and other observations indicate that purifying selection is most probably the main factor responsible for the differential accumulation of TE sequences along the *A. thaliana* genome and notably their clustering in pericentromeric regions. Finally, we reveal the importance of the mobilome as a generator of large-effect alleles at loci underlying adaptive traits. Collectively, our approaches and findings provide a unique framework for detailed studies of the dynamics and impact of transposition in nature.

## Results

### Composition of the *A. thaliana* mobilome

The reference genome sequence of *A.thaliana* is 125 Mb long (TAIR 10) and contains ~32000, mostly degenerate TE copies that belong to 326 distinct families (*Arabidopsis Genome Initiative, 2000*; *Ahmed et al., 2011*). So far, transposition activity has been documented experimentally for nine TE families, mainly on the basis of studies carried out in the reference accession Col-0 (*Ito and Kakutani, 2014*; *Tsay et al., 1993*). To assess species-wide the composition of the *A.thaliana* mobilome, we used publically available Illumina short genome sequence reads (*Schmitz et al., 2013*;

*Schneeberger et al., 2011*). First, we looked for TE copy number variation (CNV) between the reference accession Columbia (Col-0) and 211 accessions taken from across the globe. To limit the problem posed by the presence of TEs in multiple copies across the genome, with varying degrees of similarity to each other, we performed an aggregated CNV analysis based on the 11,851 annotated Col-0 TE sequences longer than 300 bp (see 'Materials and methods'). CNVs were detected for 263 TE families (*Figure 1A and B*; *Figure 1—source data 1*; see 'Materials and methods'), in keeping with the results of a previous study indicating that the vast majority of the TE sequences annotated in the Col-0 reference genome are absent from that of at least one of 80 accessions analyzed (*Cao et al., 2011*).

Since CNVs could reflect either recent TE mobilization or the gain or loss of TE copies through other types of chromosomal rearrangements, we then looked among the unmapped Illumina short reads for so-called 'split-reads' that contain TE extremities. Crucially, because most TE families generate short target site duplications (TSDs) of fixed size upon insertion, TSDs can serve as signatures of *bona fide* transposition events. We therefore developed a pipeline for the systematic identification of split-reads covering TE junctions that are absent from the reference genome and that produce, when mapped to the insertion site, a sequence overlap of the size of TSDs (3 to 15 bp, depending on the TE family, *Figure 1C and D*; see 'Materials and methods'). Our pipeline differs in that respect from that used in another study to detect the presence/absence of reference and non-reference TE insertions in the same set of accessions (*Stuart et al., 2016*). Results produced by our pipeline for the 292 annotated TE families that create TSDs upon transposition were verified visually to eliminate false positives (*Figure 1—figure supplement 1*; see 'Materials and methods'). Following this approach, non-reference TE insertions with TSDs were identified for 131 TE families in total (*Figure 1—source data 2*), which all also show CNV (*Figure 1—source data 1*).

Most (86%) non-reference TE insertions with TSDs are private or shared only by a few accessions and thus they typically correspond to recently derived alleles, as expected (*Figure 1E*). Moreover, recent transposition activity is only detected for between four and 66 TE families in any given accession, thus indicating large variations in the composition of the mobilome among accessions (*Figure 1F*). Nonetheless, we have probably identified most of the annotated TE families that compose the mobilome at the species level, because the number of TE families defined as mobile by the split-reads approach reaches a plateau after examining just 74 accessions (*Figure 1G*). The 53 class I *COPIA* families and the 40 class II Mutator-like (*MuDR*) families are the most mobile, as they account for 1408 and 729 of the 2835 non-reference TE insertions with TSDs identified in total, respectively (*Figure 1H* and *Figure 1—source data 2* and *3*). However, the number of non-reference insertions per accession is always small (<16) for any given family (*Figure 1—source data 2*), thus suggesting a lack of recent transposition bursts.

The ability to detect non-reference TE insertions with TSDs using split-reads is strongly dependent on read depth as well as sequence composition at the insertion site (*Figure 1—figure supplement 2A*; *Hénaff et al., 2015*). To assess the extent of this limitation, we used the assembled Ler-1 genome sequence recently obtained using PacBio long reads (see 'Materials and methods'). Although not annotated, this sequence assembly can serve to identify by whole genome comparison the Col-0 TEs flanked by TSDs that are absent from the corresponding position in the Ler-1 genome (see 'Materials and methods'). A total of 142 TEs belonging to 80 distinct families were identified in this way (*Figure 1—source data 4*), which is consistent with estimates obtained using other approaches (*Ziolkowski et al., 2009*; *Hénaff et al., 2015*). In contrast, we could detect only 78 Col-0-specific TEs with TSDs belonging to 49 TE families when using the split-reads pipeline to map Col-0 short reads onto the assembled Ler-1 genome. These results indicate therefore that the split-reads approach has a low sensitivity (*Figure 1—figure supplement 1C*; 45% false negatives and 10% false positives; False Discovery Rate: 15.3%).

To obtain an independent estimation of the composition of the mobilome, we also performed TE sequence capture (TE-capture; *Baillie et al., 2011*). Briefly, probes were designed to cover the 5' and 3' extremities of 310 TE elements belonging to 181 distinct families, including 117 of the 131 TE families identified as mobile with the split-reads approach (see 'Materials and methods'). Using genomic DNA extracted from 12 randomly chosen accessions (*Figure 1—figure supplement 2D*), we could validate by TE-capture most (87%) of the non-reference TE insertions with TSDs that were detected by the split-reads approach (*Figure 1—figure supplement 2F*; see 'Materials and methods'). As expected, TE-capture also uncovered many additional non-reference TE insertions with

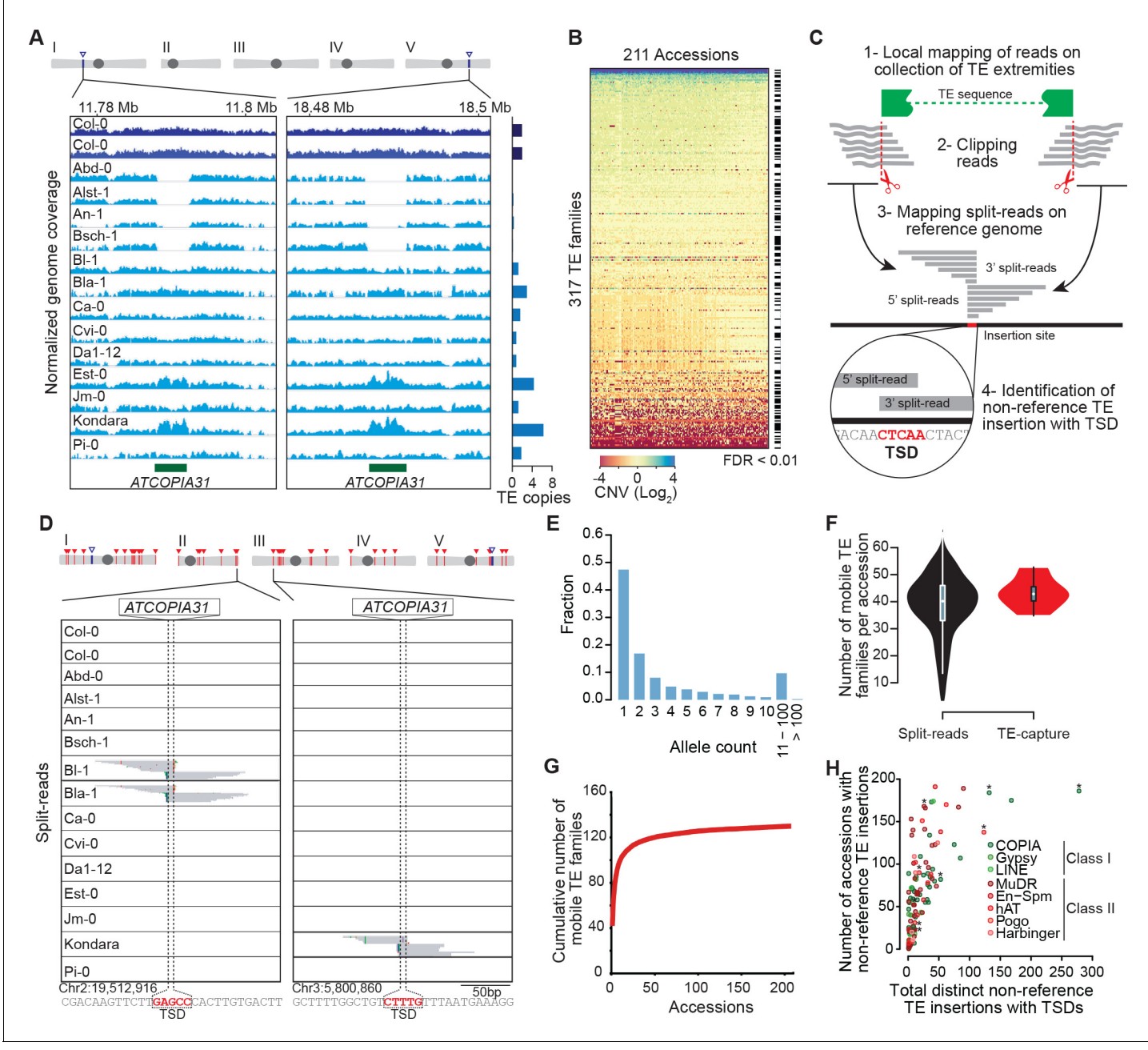

**Figure 1.** Overview of the *A. thaliana* mobilome. (A) Genome browser tracks showing normalized sequencing coverage over the two full-length *ATCOPIA31* elements annotated in the reference genome (Col-0). CNV is detected as increased or decreased coverage in other accessions. Number of copies is indicated on the right. (B) Heat map representing CNVs (log2 ratio) for 317 TE families and 211 *A. thaliana* accessions. TE families with statistically significant CNV in at least one accession are indicated. *Figure 1—source data 1* contains absolute copy number estimation of TE sequences. (C) Schematic representation of the bioinformatics pipeline to identify non-reference TE insertions with TSD using split-reads. 1- Reads are mapped on a collection of TE extremities from annotated TE sequences and reference sequences (Repbase update). 2- Reads aligning partially over TE extremities are extracted and clipped. 3- The unmapped portion of these split-reads are re-mapped on the Arabidopsis reference genome. 4- Non-reference TE insertions with TSDs are identified by searching for overlapping clusters of 5' and 3' split-reads. (D) Genome browser tracks showing split-reads for two non-reference *ATCOPIA31* insertions and TSD reconstruction. *Figure 1—source data 2* contains the coordinates of all non-reference TE insertions with TSDs. (E) Distribution frequency of allele counts for non-reference TE insertions with TSDs. (F) Number of mobile TE families per accession identified using split-read and TE-sequence capture. (G) Cumulative plot of the number of mobile TE families detected with increasing numbers of accessions. (H) The total number of non-reference TE insertions with TSDs is indicated in relation to the number of accessions with such insertions, for each of the 131 mobile TE families. Asterisks indicate the nine TE families with experimental evidence of transposition (*Ito and Kakutani, 2014; Tsay et al., 1993*). *Figure 1—source data 3* contains the total number of distinct non-reference TE insertions with TSD for each TE family and

*Figure 1 continued on next page*

*Figure 1 continued*

super-family. *Figure 1—figure supplement 2* shows TE-capture results. *Figure 1—figure supplement 1* contains IGV screenshots showing the pattern of split-reads characteristic of true- and false-positive non-reference TE insertions with TSDs.

The following source data and figure supplements are available for figure 1:

**Source data 1.** Copy number estimation of TE sequences.
**Source data 2.** Coordinates of non-reference TE insertions with TSDs.
**Source data 3.** Number of distinct non-reference TE insertions with TSDs identified by the split-reads approach for each TE family and super-family.
**Source data 4.** TE insertions with TSDs present in Col-0 but absent in Ler-1.
**Figure supplement 1.** Visual inspection of true- and false-positive non-reference TE insertions with TSDs.
**Figure supplement 2.** Validation of the *A. thaliana* mobilome by TE-capture.

TSDs for the same TE families (*Figure 1—figure supplement 2F*). However, no such insertions were detected for the other TE families that could be captured but which were not identified as mobile by the split-reads approach in any of the 12 accessions. These results confirm that despite the low sensitivity of the latter, we have probably identified most of the TE families with TSDs that compose the *A. thaliana* mobilome at the species level. Finally, non-reference insertions were also identified for 30 TE families (including 15 *HELITRON* families) that could not be analyzed using our split-reads pipeline because they do not produce TSDs or have insertion sites located in low complexity regions (*Figure 1—figure supplement 2B*). Since most of the non-reference insertions for these 30 TE families are present in only one or two of the 12 accessions examined (*Figure 1—figure supplement 2G*), they likely reflect recent transposition events. Thus, there are altogether at least 165 TE families with recent transposition activity at the species level. Moreover, based on the TE-capture data, we can estimate that since they diverged from each other any two accessions have accumulated between ~200 and ~300 newly transposed TE copies (*Figure 1—figure supplement 2H*).

## TE mobilization as a complex trait

The observation that the composition of the mobilome differs extensively between accessions (*Figure 1F*) suggests that it is influenced by environmental and genetic factors. To try to identify such factors, we first established that copy number (CN) correlates positively with the number of TE insertions with TSDs that are detected by TE-capture (*Figure 2—figure supplement 1*; see 'Materials and methods'). Thus, CNV is a reliable and quantitative estimator of differential TE mobilization between accessions, which we used to analyze the 113 TE families that were defined as mobile based both on the split-reads approach and TE-capture (*Figure 2—source data 1*).

Controlling for population stratification and considering thirteen geo-climatic variables (*Hancock et al., 2011*), we uncovered robust correlations with CN for 15 class I and class II TE families. Among these, *ATCOPIA2* and *ATCOPIA78* share the highest number of geo-climatic variables correlated with CN (*Figure 2—figure supplement 2*). Moreover, the strongest correlation is between temperature annual range and CN for *ATCOPIA78* (*Figure 2A and B*). Given that at least one member of this TE family is transcriptionally induced by heat shock in the Col-0 accession (*Ito et al., 2011*; *Cavrak et al., 2014*), *ATCOPIA78* provides a compelling case of a causal link between climate and TE mobilization.

We next explored the possibility of using GWASs to identify genetic variants influencing TE mobilization (see 'Materials and methods'). For 33 TE families, a disproportionately large number of SNPs are associated with CNV, preventing further analysis. For the remaining 80 TE families, SNPs in linkage disequilibrium with each other and associated with CNVs delineate 230 loci. Moreover, 34% of these loci are also identified by GWAS using whole genome sequencing data obtained for another 180 accessions taken from Sweden (*Long et al., 2013*) (*Figure 2—figure supplement 3A*). This substantial overlap suggests a similar genetic architecture for the *A. thaliana* mobilome at both the local

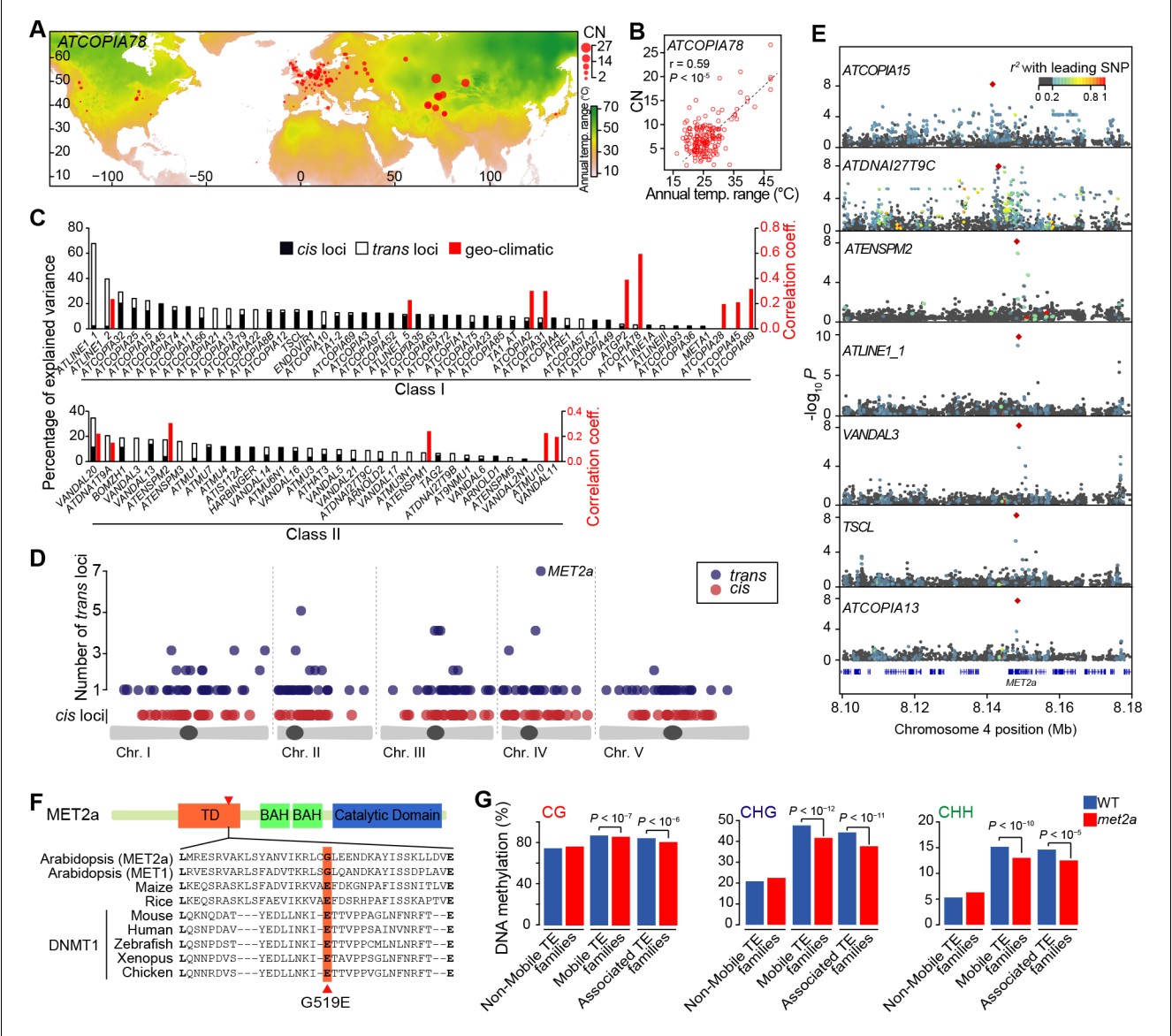

**Figure 2.** Environmental and genetic factors associated with differential mobilome activity. (**A**) Copy number (CN, red circles) of *ATCOPIA78* in accessions distributed across the globe. Annual temperature range is also shown. (**B**) Partial Mantel correlation between *ATCOPIA78* CN and annual temperature range. (**C**) Fraction of CNV variance explained by SNPs (*cis*, and *trans*) and partial Mantel correlation with geo-climatic variables. (**D**) Distribution of *cis* and *trans* loci in the joined analysis (391 accessions) and number of TE families associated with a given *trans* locus. A complete list of the GWAS results is provided in *Supplementary file 1*. (**E**) Manhattan plots displaying GWAS results for the seven TE families with a *MET2a* association. The leading SNP within each interval is indicated as a red diamond. Colors indicate the extent of linkage disequilibrium ($r^2$) with the leading SNP. (**F**) Schematic view of the MET2a protein (TD: targeting domain; BAH: bromo adjacent homology domain) and sequence alignment of the TD. The amino acid substitution (G519E) that is present in some accessions is indicated (red arrow). (**G**) Average DNA methylation level over non-mobile, mobile and *MET2a*-associated TE families in WT and *met2a* Col-0 seedlings (*Stroud et al., 2013*). Statistically significant differences are indicated (MWU test). *Figure 2—figure supplement 1* shows the positive correlation between CN and number of non-reference TE insertions with TSDs. *Figure 2—figure supplement 2* shows climate association to CNVs. *Figure 2—figure supplement 3* shows GWAS results for CNVs.

The following source data and figure supplements are available for figure 2:

**Source data 1.** Copy number estimation used for the geo-climatic associations and GWASs.

**Figure supplement 1.** Pearson correlation between TE CN and number of TE sequences identified by TE capture.

**Figure supplement 2.** Climate association to TE CNV.

*Figure 2 continued on next page*

*Figure 2 continued*

**Figure supplement 3.** GWAS of CNVs.

and global scales, which prompted us to perform joined GWASs using all 391 accessions in order to increase both sensitivity and specificity. Depending on the TE family, GWASs identified from 0 to 33 loci and collectively, associations explain between 2% and 67% of the variance in CN (*Figure 2C*).

Among the 334 loci detected in total, 130 encompass sites with reference or non-reference TE insertions (*Figure 2C* and *Supplementary file 1*). Furthermore, each of these local (*cis*) genetic variants explains on average 5.2% of the total variance compared with 2.2% for distal (*trans*) genetic variants. The higher explanatory power of *cis* variants is of course to be expected, as the TEs themselves are the primary determinants of the transposition process. Indeed, almost all *cis* SNPs that map to TE sequences in the reference genome are likely causal as they affect sequences involved in transposition, for example the long terminal repeats (LTRs) and the various open reading frames of LTR retrotransposons (*Figure 2—figure supplement 3B and C*).

While *cis* loci collectively explain most CN variance for class I TE families, this is not the case for class II TE families (*Figure 2—figure supplement 3D*). Given that class II TEs move by a cut and paste mechanism, some *trans* loci could in fact correspond to sites of excision. However, we could not find evidence of excision footprints, such as small insertions or deletions. Alternatively, the larger fraction of CN variance explained by *trans* loci for class II TE families may in part result from many of these families being non-autonomous, i.e. requiring factor(s) encoded by other TEs for their mobilization. Consistent with this possibility, the proportion of CN variance associated with *trans* loci as well as the number of TE annotations overlapping *trans* loci are higher for non-autonomous than autonomous class II TE families (*Figure 2—figure supplement 3E*). Although we did not investigate *trans* mobilization in depth, we readily identified one probable case, involving the non-autonomous and autonomous MuDR families *ATDNA1T9A* and *VANDAL16*, respectively (*Figure 2—figure supplement 3F*). Of note, CN do not co-vary between these two families, which could indicate that their transposition is differentially controlled. Finally, 'false' *trans* loci could also be caused by non-reference TE insertions that are sufficiently frequent to be in linkage disequilibrium with SNPs used for the GWASs but that we have failed to detect. However, such *trans* loci are not expected to be more prevalent for class II than class I TE families and should be very rare in any case given that the probability of missing moderately frequent (>5%) non-reference TE insertions by both the split-reads approach and TE-capture is low (*Figure 2—figure supplement 3G*).

Since transposition is controlled by multiple protein activities in addition to those encoded by the TEs themselves, we also examined genes located within or adjacent to *trans* loci. Overall, these genes do not appear to be enriched for any particular function and most of them are specific to a single TE family (*Figure 2D* and *Figure 2—figure supplement 3H*). These observations indicate either a complex genetic architecture of mobilome variation and/or spurious *trans* associations such as those considered above. Nonetheless, 22 *trans* loci stand out as they show association with CNV for two or more TE families (*Figure 2D*) and a causal link is evident in two cases. Indeed, the locus associated with CNV for respectively the retrotransposon and DNA transposon families *ATGP2* and *ATENSPM2* encodes the transcription factor ARF23, which recognizes motifs that are overrepresented in the sequence of these TEs. The second locus is associated with CNV for the largest number of TE families (four class I and three class II families, *Figure 2E*) and encodes the MET2a protein, a poorly characterized homolog of the main DNA maintenance methyltransferase MET1. Moreover, one of the *MET2a* SNPs is presumably causal as it leads to a non-synonymous amino-acid substitution (G519E) in a conserved domain of the protein (*Figure 2F*) that in the mammalian homolog Dnmt1 is required for the targeting to replication foci (*Klein et al., 2011*). A role for *MET2a* is also supported by the observation that *met2a* mutant plants (*Stroud et al., 2013*) lose some DNA methylation exclusively over mobile TE families. Furthermore, loss of methylation is more pronounced when only considering the seven TE families that show a *MET2a* association (*Figure 2G*). Intriguingly, CHG sites (where H=A, T or C), which are poor substrates for MET1 or Dnmt1 compared to CG sites (*Law and Jacobsen, 2010*), are the most affected in the *met2a* mutant. Whether or not this

observation reflects an atypical recognition specificity for MET2a remains to be determined. Finally, we note that GWASs failed to detect any association with genes known to be involved in the epigenetic silencing of TEs (*Ito and Kakutani, 2014*) such as *MET1* and *DDM1*, presumably because of their essential function.

## Genome localization of newly inserted TEs

In *A. thaliana* as in many other eukaryotes, TE sequences tend to cluster in pericentromeric regions (*Arabidopsis Genome Initiative, 2000*). Mechanistically, such clustering may result from insertion bias, selective constraints or differential elimination of TE copies through ectopic homologous recombination (*Barrón et al., 2014*). To distinguish between these possibilities, we looked at the genomic location of the 2835 non-reference TE insertions with TSDs detected with the split-reads approach and found that they are distributed almost evenly along chromosomes (*Figure 3A*). Since a similar distribution is observed for the non-reference TE insertions with TSDs detected exclusively using TE capture (*Figure 3—figure supplement 1A*), we can rule out an ascertainment bias of the split-reads approach towards non-reference TE insertions located along the chromosome arms. However, there is a clear trend towards a more pericentromeric localization when only considering non-reference TE insertions with TSDs that are shared by two or more accessions and that are thus presumably more ancestral (*Figure 3B* and *Figure 3—figure supplement 1B*). Moreover, the density of non-reference TE insertions with TSDs positively correlates with the recombination rate but negatively with gene density (*Figure 3—figure supplement 1C, D and E*). Finally, except for *COPIA* families, non-reference TE insertions with TSDs are globally under-represented within genes, where they are expected to be most detrimental (*Figure 3C* and *Figure 3—figure supplement 1F* and *Figure 3—figure supplement 2A*). Collectively, these observations provide strong evidence that TEs insert equally throughout the genome and are preferentially purged over time from the chromosome arms because of their deleterious effects on adjacent genes rather than as a consequence of ectopic homologous recombination.

Although most TE families show no overt insertion bias at the genome scale, there are clear local insertion preferences. In agreement with previous observations (*Fu et al., 2013*; *Miyao et al., 2003*), private non-reference *COPIA* and *MuDR* insertions with TSDs are enriched at coding sequences and transcriptional start sites (TSS), respectively (*Figure 3—figure supplement 2A*). In addition, insertion sites for most TE superfamilies are enriched in specific DNA sequence motifs or exhibit biased sequence composition (*Figure 3—figure supplement 2B and C*; *Supplementary file 2*). For example, *LINEs* tend to insert within poly(A) tracks, as expected for this superfamily of non-LTR retrotransposons, which integrate into the genome via poly(A)-dependent, target site–primed reverse transcription (TPRT; *Beck et al., 2011*).

## Impact of newly inserted TEs on the expression of adjacent genes

Transcriptome analyses in the reference accession Col-0 have revealed that *A.thaliana* genes nearest to TE sequences are expressed at lower levels compared with the genome-wide distribution of gene expression, suggesting that TE insertions tend to reduce the expression of neighboring genes (*Hollister and Gaut, 2009*). To investigate more directly the impact of TEs on the genes within or near which they insert, we examined RNA-seq data available for 144 accessions (*Schmitz et al., 2013*). Specifically, we considered all non-reference TE insertions with TSDs and calculated for each gene located within 1 kb of them (1616 genes in total), the ratio between the expression level in the accession(s) harboring the insertion and the median expression level in the accessions devoid of the insertion. Expression ratios expected under the null hypothesis (no effect of the TE insertions) were calculated by taking $10^6$ randomly chosen sets of 1616 genes and assigning for each set the TE insertion 'presence/absence' label randomly among the 144 accessions (see 'Materials and methods'). Comparison of the distribution of the observed and expected expression ratios indicates that for a large fraction of genes, expression is indeed significantly altered when TEs insert within or near them (*Figure 3D*, $p < 1.9 \times 10^{-5}$). These alterations are most pronounced for the *COPIA* insertions, which are overrepresented in genes and less pronounced for the *MuDR* insertions, despite the latter being overrepresented around the TSS of genes (*Figure 3E* and *Figure 3—figure supplement 2A*). Although other TE superfamilies show similar trends, we could not draw firm conclusions in these cases because of insufficient statistical power (*Figure 3—figure supplement 3*). This

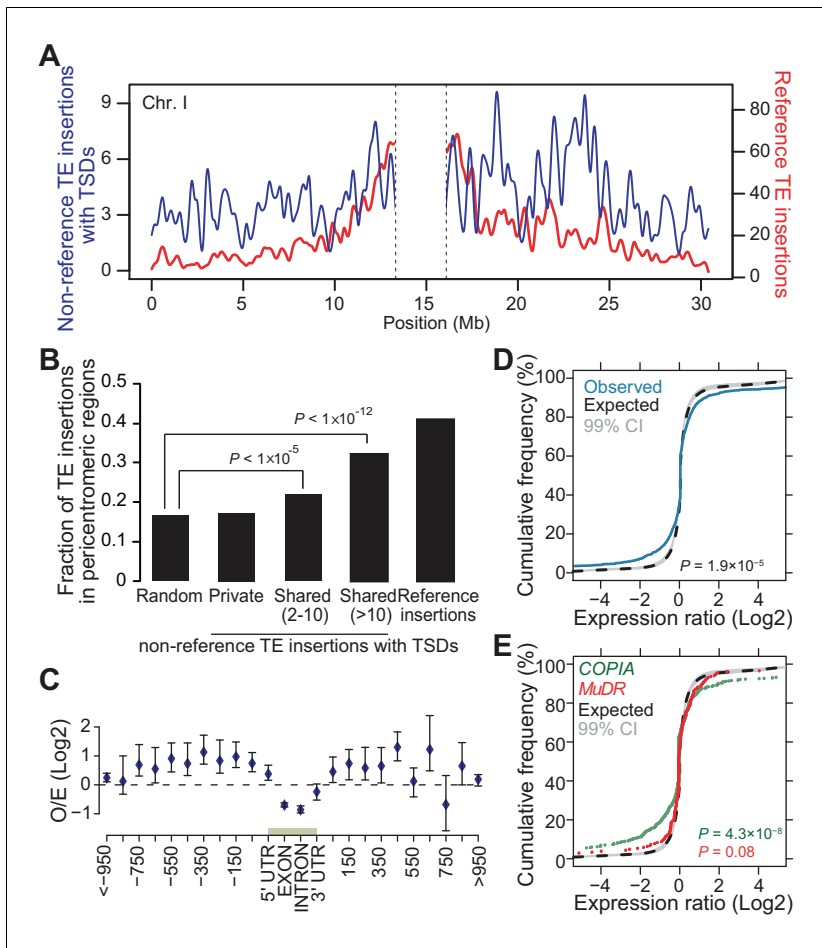

**Figure 3.** Genomic localization of non-reference TE insertions. (**A**) Density of non-reference TE insertions with TSDs (blue) and of annotated TE sequences (red) along the reference sequence of chromosome 1. Inner pericentromeric regions are masked. (**B**) Fraction of private and shared non-reference TE insertions with TSDs and of annotated TE sequences in outer pericentromeric regions. Statistically significant differences are indicated (chi square test). (**C**) Observed/expected ratio (O/E) of private non-reference TE insertions with TSDs in and around genes. Errors bars are defined as 95% confidence intervals. (**D**) Cumulative distribution of gene expression ratios between alleles harboring and lacking non-reference TE insertions. Statistically significant differences were calculated using the KS test. (**E**) As D, but only for *COPIA* (green) or *MuDR* (red) non-reference TE insertions with TSDs. *Figure 3—figure supplement 1* shows detailed analysis of the distribution of non-reference TE insertions with TSDs along the genome. *Figure 3—figure supplement 2* shows local TE insertion preferences. *Figure 3—figure supplement 3* shows global expression levels of gene affected by non-reference TE insertions. *Figure 3—figure supplement 4* shows expression levels of genes affected in some accessions by a non-reference insertion with TSD in plants grown under control conditions or subjected to heat stress.

The following figure supplements are available for figure 3:

**Figure supplement 1.** Distribution of non-reference TE insertions with TSDs along the genome.

**Figure supplement 2.** Local TE insertion preferences.

**Figure supplement 3.** Global expression levels of genes affected by non-reference TE insertions of different TE superfamilies.

**Figure supplement 4.** Expression levels of selected genes affected by non-reference TE insertions with TSDs.

notwithstanding, it is clear that TE insertions induce both increases and decreases in gene expression with equal frequency (*Figure 3D and E*). Thus our findings contradict the prevailing view of a dampening effect of TE insertions on the expression of adjacent genes (*Hollister and Gaut, 2009*) and suggest instead a stronger selection against TE insertions when they occur close to highly expressed genes.

To complement the re-analysis of transcriptome data, we also measured by RT-qPCR the expression level of 19 genes with recent *COPIA* or *MuDR* insertions, using nine different accessions grown under control conditions or subjected to heat shock. *COPIA* insertions were found to have more dramatic and systematic effects on gene expression in stressed plants (*Figure 3—figure supplement 4*) which in the case of *ATCOPIA78* can be related to its transcriptional sensitivity to heat shock (*Ito et al., 2011*; *Cavrak et al., 2014*). On the other hand, we could not detect any effect of the MuDR insertions under the two conditions tested (*Figure 3—figure supplement 4*). These findings are in agreement with those of the transcriptome analysis and indicate in addition that the effect of TE insertions on the expression of adjacent genes can vary substantially in relation to the environment.

## Impact of newly inserted TEs on the DNA methylation status of adjacent sequences

TE sequences are typically targeted by the RNA-directed DNA methylation (RdDM) machinery in *A. thaliana* (*Lippman et al., 2004*; *Lister et al., 2008*; *Cokus et al., 2008*) and we have previously provided genome-wide evidence that DNA methylation can spread from RdDM targets to flanking sequences, with possible consequences on gene expression (*Ahmed et al., 2011*). To investigate the effect of new TE insertions on the DNA methylation status of adjacent sequences, we used MethylC-Seq data available for 140 accessions (*Schmitz et al., 2013*). Analysis of this data set first indicated that mobile TE families have on average higher CG, CHG and CHH methylation than non-mobile TE families (*Figure 4A*). Furthermore, DNA methylation is also higher for most mobile TE families in the accessions with evidence of recent transposition activity (*Figure 4—figure supplement 1A*). These observations prompted us to examine in addition methylome data obtained for several mutation accumulation (MA) lines (*Becker et al., 2011*; *Schmitz et al., 2011*). Mobile TE families suffer less sporadic DNA methylation loss than non-mobile families (*Figure 4B*). These findings are entirely consistent with DNA methylation playing an important role in the control of TE mobility and they suggest in turn that most of the recent TE insertions we have identified are present in the methylated state. Moreover, given that DNA methylation is likely established over newly inserted TE copies by RdDM in a progressive manner across multiple generations (*Teixeira et al., 2009*; *Marí-Ordóñez et al., 2013*), unmethylated non-reference TE insertions should be mainly private and reflect very recent transposition events.

Based on these considerations, we next analyzed the DNA methylation status of 1543 TE insertion sites for which reliable data could be extracted across all 140 accessions (*Figure 4C*). Approximately 10% of sites are methylated in most accessions, including systematically in the one(s) containing the TE. As expected, these sites are preferentially located within TE-rich, pericentromeric regions. In contrast, another 40% of sites are devoid of methylation in the accession(s) containing the TE insertion as well as in most of the other accessions. This absence of adjacent DNA methylation could indicate either that the TE insertions themselves are unmethylated or else that DNA methylation does not spread from them. Finally, 50% of sites are methylated exclusively or almost exclusively in the accession(s) with the TE insertion (*Figure 4—figure supplement 1B*), thus suggesting that at these sites TEs are methylated and that DNA methylation did spread into adjacent sequences. Why some sites may be refractory to DNA methylation spreading when others are not is unclear, as we did not identify any feature that could distinguish them, such as the identity of the TE or the sequence composition at the insertion site.

Further analysis of DNA methylation associated with TE insertions indicates that it affects all three sequence contexts and that it generally extends for up to 300 bp on both sides of the insertions (*Figure 4D and E*; *Figure 4—figure supplement 1B*), a distance that closely matches that previously reported for the spreading of DNA methylation from RdDM targets (*Ahmed et al., 2011*). For 243 insertion sites however, DNA methylation extends over much longer distances (up to 3.5 kb) on one or the other side of the insertion (*Figure 4D and E*). While most of these sites lie within or close to genes, the TE insertions are not preferentially orientated with respect to gene transcription

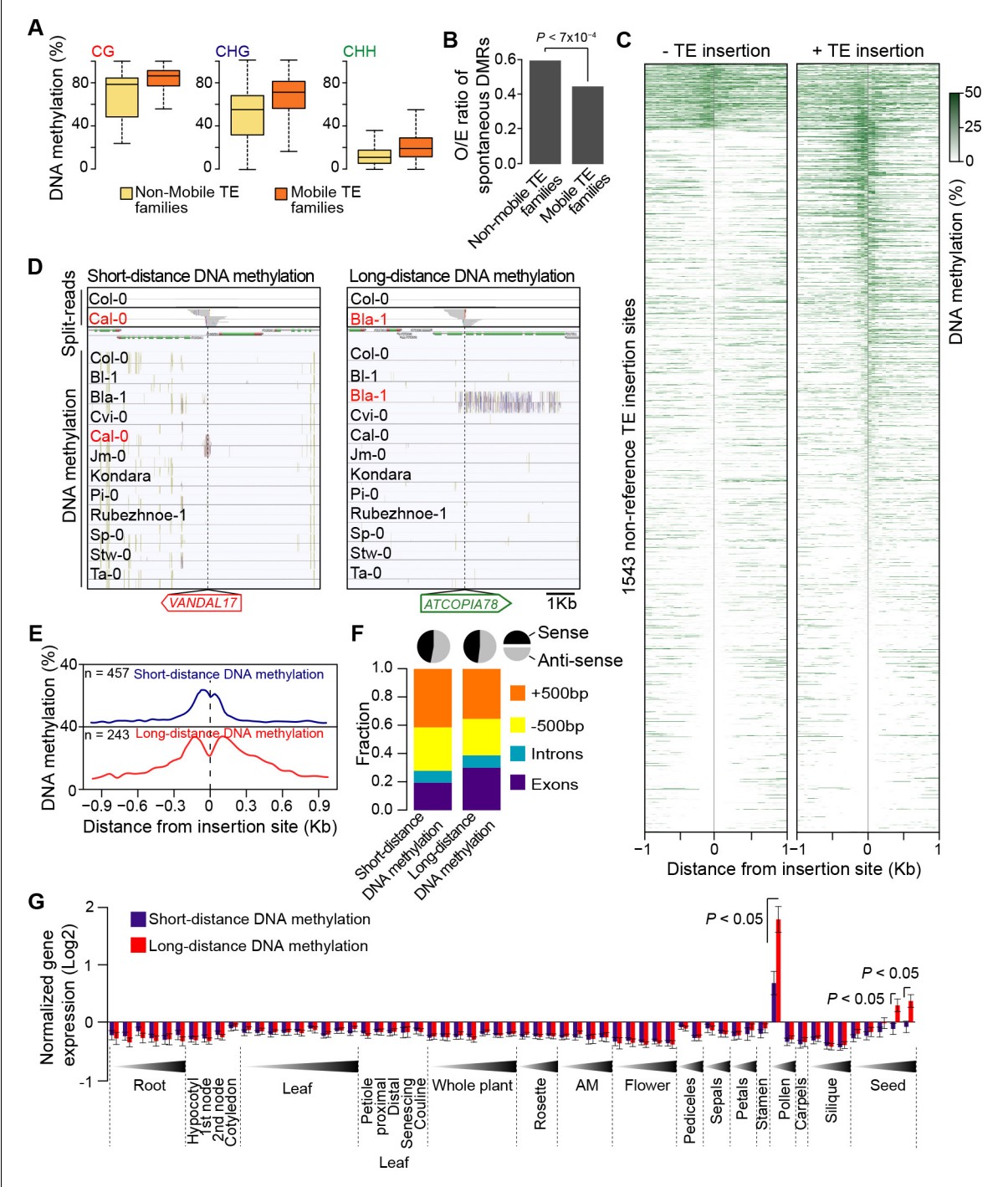

**Figure 4.** DNA methylation of non-reference TE insertion sites. (A) Boxplot representation of average DNA methylation level for mobile and non-mobile TE families across all accessions. (B) O/E ratio of spontaneous DMRs identified in mutation accumulation lines (*Becker et al., 2011*; *Schmitz et al., 2011*) for non-mobile and mobile TE families. Statistically significant differences were calculated using a chi square test. (C) Average DNA methylation level in 50bp windows upstream and downstream of 1543 insertions sites for accessions lacking or containing a given non-reference TE insertion with TSD. (D) Genome browser tracks showing examples of insertion sites respectively associated with short- and long-distance DNA methylation. (E) Meta-analysis of DNA methylation around non-reference TE insertions sites. (F) Distribution of non-reference TE insertions associated with short- or long-distance DNA methylation according to their position relative to genes (stacked bar plot) and proportion of insertions in the two possible orientations relative to the closest gene (pie charts). (G) Average expression level in different organs and at different developmental time points (in Col-0) of genes with non-reference TE insertions with TSDs and affected by short- (blue) or long-distance (red) DNA methylation. Error bars

*Figure 4 continued on next page*

*Figure 4 continued*

are s.e.m. Statistical significance of differences was calculated using a MWU test. *Figure 4—figure supplement 1* shows DNA methylation of TE families and impact on sequences flanking non-reference TE insertions with TSDs.
The following figure supplement is available for figure 4:

**Figure supplement 1.** DNA methylation of TE families and impact on sequences flanking non-reference TE insertions with TSDs.

(*Figure 4F*), which rules out sense-antisense transcription as the likely trigger for this long-distance DNA methylation. Proximity to pericentromeric heterochromatin can also be ruled out, because most of the genes with long-distance DNA methylation are located on the chromosome arms (*Figure 4—figure supplement 1C*). To explore potential mechanisms further, we made used of the wealth of epigenomic data available in Col-0 to examine the 142 Col-0 TE insertions with TSDs that are absent from the assembled Ler-1 genome sequence (*Figure 1—source data 4*). Methylome data (*Stroud et al., 2013*) indicate that 121 of these 142 Col-0 TE copies are methylated and that DNA methylation tends to extend into flanking sequences, predominantly over short distances, but occasionally over much longer distances (<300 pb: 63 TE insertions; >1 kb: 36 TE insertions; *Figure 1—source data 4*). These results confirm those obtained for the non-reference TE insertions. In addition, analysis of Col-0 small RNA-seq data (*Fahlgren et al., 2009*) indicates that in contrast to short-distance DNA methylation, long-distance DNA methylation aligns with 24-nt siRNAs (*Figure 4—figure supplement 1D*). Thus, genes affected by the latter type of DNA methylation have presumably become secondary targets of RdDM, as was shown for a transgene (*Kanno et al., 2008*; *Daxinger et al., 2009*). Moreover, genes affected by long-distance DNA methylation in accessions other than Col-0 tend in the latter accession, where they are by definition in the ancestral state, to be most highly expressed in both pollen and seeds and more highly expressed in these two organs than genes affected by short-distance DNA methylation (*Figure 4G*). Given that RdDM activity is also maximal in these organs (*Teixeira and Colot, 2010*), our observations suggest that secondary RdDM results from the concomitance of strong transcription and strong RdDM at target loci.

Finally, our analysis of TE-associated DNA methylation indicates that it accounts for at least 7% of the so-called gene C-DMRs (i.e. regions of differential methylation at CG, CHG and CHH sites) identified in nature, which are typically low frequency gain of DNA methylation variants (*Schmitz et al., 2013*). These and similar findings reported recently (*Stuart et al., 2016*) confirm and extend previous results that first indicated that many natural gene C-DMRs are not *bona fide* epialleles but rather new alleles caused by TE insertions (*Schmitz et al., 2013*). Nonetheless, examination of one TE-insertion allele shared among 13 accessions indicates that it is present in the unmethylated state in one accession and thus possibly subjected to epigenetic variation in nature (*Figure 4—figure supplement 1B*).

## TE insertions as motors of adaptive changes

Although TEs tend to insert with no overt bias at the genome scale (*Figure 3A*), we detected nineteen 10 kb windows with a high load of non-reference TE insertions (*Figure 5A*). Such enrichment could result from insertion preferences or reflect an absence of strong negative selection. In fact, three of these 10 kb windows span genes encoding nucleotide-binding domain and leucine-rich repeat containing (NLR) proteins, which function as immune receptors in plants and are known to be under diversifying selection (*Chae et al., 2014*). Moreover, a fourth 10 kb window spans the gene *FLC*, which encodes a key repressor of flowering and is one of the main genetic factors causing natural variation in the onset of flowering, another key adaptive trait (*Ietswaart et al., 2012*). Remarkably, the *FLC* locus has the highest number of non-reference TE insertions (seven in total) across the genome. These insertions belong to several COPIA families and affect four distinct *FLC* haplotypes in total (*Figure 5A* and *Figure 5—figure supplement 1A*). Moreover, five insertions are located within the first intron (*Figure 5B*), which plays an important role in the epigenetic regulation of *FLC* in response to cold (*Ietswaart et al., 2012*). Although four of these insertions as well as another intronic insertion were previously identified among early flowering accessions (*Liu et al., 2004*; *Lempe et al., 2005*), causality could not be established unequivocally because of numerous other

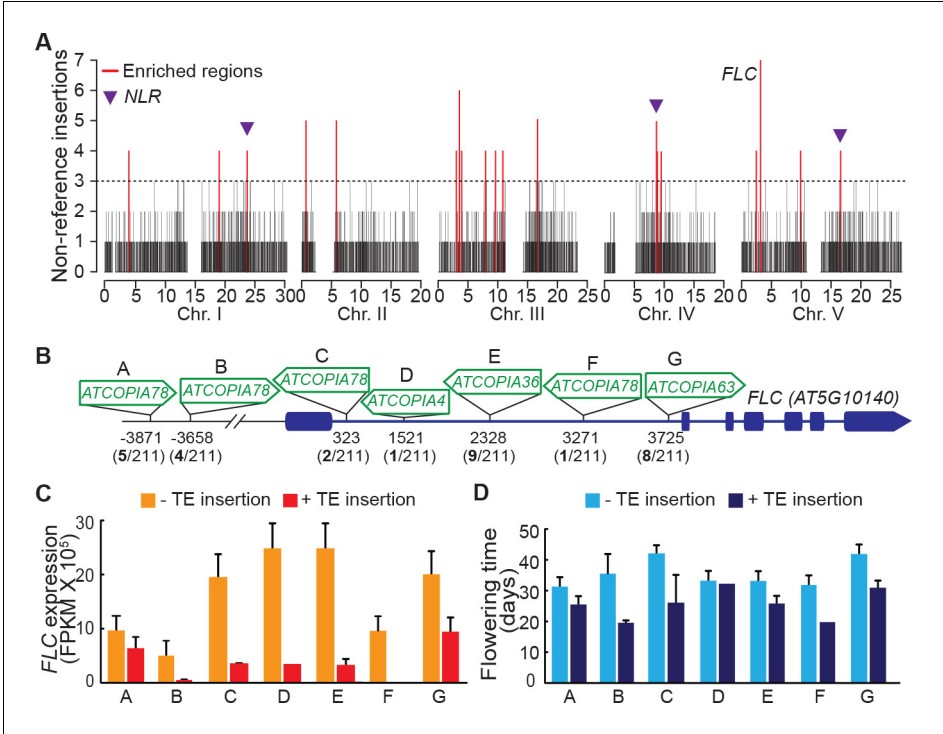

**Figure 5.** Local enrichment of non-reference TE insertions with TSDs. (**A**) Density of non-reference TE insertions with TSDs in 10 kb windows. The 19 regions statistically enriched in such insertions are indicated by red bars. (**B**) Position and identity of the seven non-reference TE insertions with TSDs spanning the *FLC* locus. (**D**) and (**E**) Level of *FLC* expression (**D**) and flowering time (**E**) for accessions of same *FLC* haplotype but differing by the presence or absence of the relevant TE insertion. Errors bars are s.e.m. *Figure 5—figure supplement 1* shows the reconstruction of the *FLC* haplotypes and additional analyses of the effect on flowering timeof non-reference TE insertions with TSDs at the locus.

The following figure supplement is available for figure 5:

**Figure supplement 1.** Reconstruction of the *FLC* haplotypes and additional analyses of the effect on flowering time of non-reference TE insertions with TSDs at the locus.

---

sequence polymorphisms in complete linkage disequilibrium. To obtain direct proof of a causal role for the seven TE insertions we identified, we used publically available transcriptomic (*Schmitz et al., 2013*) as well as phenotypic data (*Li et al., 2010*; *Lempe et al., 2005*) and compared *FLC* expression as well as flowering time among accessions that have the same *FLC* haplotype but differ by the presence or absence of a TE insertion (see 'Material and methods'). Results of these comparisons indicate that the TE-containing accessions have systematically much reduced *FLC* expression and flower much earlier than their TE-free counterparts (*Figure 5C and D*; *Figure 5—figure supplement 1B and C*). Thus, we can conclude that TEs are recurrent generators of major effect *FLC* alleles, which in turn suggests that they contribute significantly to the high level of allelic heterogeneity observed at this locus (*Li et al., 2014*)

## Discussion

We have shown that the *A. thaliana* mobilome is particularly rich at the species level, being composed of at least half of the 326 TE families that are annotated in the reference genome. This finding is at odds with the prevailing view that most TE families are mere molecular fossils in *A. thaliana*, since they contain a much lower proportion of 'young', i.e. non-degenerated TE copies than in the close relative *A. lyrata* (*Hu et al., 2011*; *Maumus and Quesneville, 2014*). Furthermore, we provide definitive evidence that TEs insert throughout the genome, with no overt bias towards

pericentromeric regions, which contrasts with the observed clustering of annotated TE sequences around centromeres. However, these discrepancies are easily resolved, since we have also shown that despite the richness of the *A. thaliana* mobilome, most TEs tend to be rapidly purged by natural selection in this species when they insert in the chromosome arms, which are gene-rich. Indeed, our systematic survey indicates that TEs have pervasive effects on the expression and DNA methylation status of genes near or within which they insert. Incidentally, the deleterious effects associated with most transposition events in *A. thaliana* may also explain in part the fact that we did not detect any recent transposition bursts, as these should be strongly counter-selected. Furthermore, because *A. thaliana* is predominantly self-fertilizing, the purging of deleterious TE insertions should be accelerated in this species compared to *A. lyrata*, which is an obligated out-crosser. Given this difference in mating systems, the TE population dynamics of *A. thaliana* and *A. lyrata* are expected to differ significantly (*Lockton and Gaut, 2010*). Thus, homologous recombination could play a more prominent role in the elimination of TE insertions in *A. lyrata,* as is thought to be the case in *D. melanogaster* (*Barrón et al., 2014*). However, comprehensive studies similar to those presented here remain to be performed for *A. lyrata* in order to identify conclusively the forces that shape the TE landscape in this species.

We have also shown that the composition and activity of the mobilome vary greatly between accessions. GWASs revealed that this variation is caused in part by sequence polymorphisms within the TEs themselves (*cis* variation), which is in agreement with empirical data and theoretical models indicating that TE families contain only one or a few active, autonomous (i.e. master) TE copies at any one time (*Becker et al., 2011*). The fact that we could readily detect such *cis* variants may again be linked to the mating system of *A. thaliana*, which on the one hand should increase the probability that disabling mutations accumulate within the few active TE copies that are present within a given lineage, before these copies could transpose further; and on the other hand should decrease the probability of acquiring new active copies through crosses.

Another important result of the GWASs is that natural variation at the *MET2a* locus, which encodes a poorly characterized DNA methyltransferase, has a significant impact on mobilome composition and activity across accessions, being associated with differential transposition activity for seven class I and class II TE families. While the role of MET2a in transposition control remains to be determined experimentally, it is noteworthy that none of the epigenetic repressors of TE activity identified through genetic screens, such as MET1 or DDM1, are associated with natural variation of the mobilome, presumably because of their essential function. Altogether, these findings illustrate the power of GWASs in identifying the genetic factors affecting transposition in nature.

Being a complex trait, TE mobilization is also modulated by environmental factors, and we have identified temperature annual range as a clear contributor to the variation in *ATCOPIA78* mobilization across accessions. Remarkably, this TE family has generated several rare alleles with large effects at the *FLC* locus, which is a major genetic determinant of the onset of flowering in nature. It is therefore tempting to speculate that *ATCOPIA78* may endow *A. thaliana* with a unique ability to adapt to global warming and the associated increase in droughts by facilitating the creation of early flowering *FLC* alleles. Additionally, these observations may provide insights into how *A. thaliana* has been able to colonize efficiently the entire northern hemisphere from a few glacial refugia located in Southern Europe (*François et al., 2008*).

In summary, our findings have far reaching implications, as they indicate that part of the missing heritability that plagues many GWASs may be accounted for by recent and thus rare TE insertion alleles with large effects (*Vinkhuyzen et al., 2013*; *Brachi et al., 2011*). More generally, our study highlights the need for similar species-wide explorations of the mobilome in a variety of organisms in order to assess the true mutational and epimutational impact of transposition as well as its contribution to natural phenotypic variation. In this respect, it can be anticipated that the advent of long read sequencing technologies will greatly facilitate such studies, especially in organisms with large, repeat-rich genomes.

# Materials and methods

## Data sources

Whole genome sequencing data (Illumina short paired-end reads) for *A. thaliana* accessions were obtained from the NCBI SRA archive for 210 accessions collected worldwide (*Schmitz et al., 2013*) SRA012474 as well as for the 180 Swedish accessions (*Long et al., 2013*; SRA052536). Illumina short paired-end reads from Ler-1 (*Schneeberger et al., 2011*) and the assembled Ler-1genome obtained using PacBio long reads (unpublished) were retrieved from http://1001genomes.org/data/MPI/MPISchneeberger2011/releases/current/Ler-1/Reads/ and https:// github. com/ PacificBiosciences/ DevNet/ wiki/ Arabidopsis-P5C3, respectively. High quality SNP imputations for these accessions were obtained from the 1001 Genomes Project (http://1001genomes.org/). Processed MethylC-seq and RNA-seq data from accessions collected worldwide (*Schmitz et al., 2013*) were obtained from NCBI GEO (GSE43857 and GSE43858, respectively). Processed BS-seq data for the *met2a* mutant (*Stroud et al., 2013*) were obtained from NCBI GEO (GSE39901). Flowering time data were retrieved from the phenotypic information deposited at https://www.arabidopsis.org/servlets/TairObject?type=germplasm&id=6530472136 as well as (*Li et al., 2010*; *Lempe et al., 2005*).

Local climate data and coordinates for the 211 accessions analyzed using CNV were retrieved from (*Hancock et al., 2011*) (http://bergelson.uchicago.edu/wp-content/uploads/2015/04/allvars948_notnormd_011311.txt.zip). TE-capture sequencing data have been deposited in the ENA short read archive under project number PRJEB11706 and accessions codes ERR1121179 to ERR1121190.

## CNV analysis

We performed an aggregated CNV analysis for each TE family by considering only annotated TE sequences longer than 300bp as a joined pseudo-annotation. Families composed in the Col-0 reference genome exclusively of TE sequences shorter than 350bp (seven families, *AR12*, *AR3*, *ATCLUST1*, *ATDNATA1*, *ATMSAT1*, *ATSINE1* and *ATSINE3*), that contain too many (>1000) copies or that overlap with genomic regions with abnormal coverage (four families, *ATREP15*, *ATREP10D*, *ATREP3* and *HELITRONY3*; see next section) were not considered. Illumina pair-end short reads were mapped onto the TAIR10 reference genome using Bowtie2 (using the arguments –mp 13 –rdg 8,5 –rfg 8,5 –very-sensitive) and PCR-duplicates were removed using Picard. Read coverage (RC) was computed in non-overlapping windows of 100bp spanning the joined pseudo-annotation for each TE family. RC over each bin was corrected for GC content (*Yoon et al., 2009*) and normalized to the genome-wide RD. Normalized RC across joined pseudo-annotations were compared between each accession and two independent re-sequenced Col-0 reference genomes (*Schmitz et al., 2013*; *Long et al., 2013*). CNVs were detected by performing a distribution-independent permutation test for each of the two references. In order to be as stringent as possible, the maximum *P*-value from these two comparisons were considered. These p-values were compared with an empirical null-distribution of p-values constructed by randomizing a million times the TE annotation labels over the 100bp windows. We defined the false discovery rate (FDR) of an observed *P*-value as the fraction of significant (that is, below the observed *P*-value in question) hits found in the randomized set. We considered statistically significant TE-CNVs when FDR was below $10^{-5}$.

## Filtering out genomic regions with aberrant coverage or low sequence complexity

Read mapping to the reference *A. thaliana* genome sequence revealed several regions with very high coverage, which correspond mainly to the 180 bp centromeric repeat unit, the 45S and 5S rDNA repeat units, *ATHILA* TE sequences, telomeres and plastid DNA-like sequences. These as well as the few regions not suitable for sequence alignment because of low complexity were identified by mapping genome sequencing reads obtained for two independent lines of the Col-0 accession (*Schmitz et al., 2013*; *Long et al., 2013*) onto the TAIR10 genome sequence. RC was calculated on consecutive non-overlapping windows of 100 bp across the entire genome. After correction for GC content (*Yoon et al., 2009*), consecutive windows (allowing one window gap) with a RC greater or lower than three median absolute deviations from the median RC signal were merged to define

larger segments. These sequences spanned 1,125,487 bp (0.94% of the TAIR10 genome sequence) and were excluded from all analyses.

## Identification of non-reference TE insertions with TSDs using split-reads

The split-read analysis was performed in four steps: (i) extraction of reads not mapping to the reference genome; (ii) forced mapping to a collection of TE sequence extremities and soft-clipping of mapped reads; (iii) mapping of clipped reads to the reference genome; (iv) identification of clipped reads that reveal target site duplications (TSDs). Briefly, for each accession we retrieved reads that do not map to the TAIR10 reference genome (containing the SAM flag 4), but we did not make use of information about discordant paired-end reads, unlike in the Jitterbug approach (*Hénaff et al., 2015*). Unmapped reads were then aligned (using Bowtie2 in –local mode to allow for soft clip alignments) to a collection of 5' and 3' TE sequence extremities (300 bp) obtained from the Col-0 for each TE family or from Repbase Update (*Jurka et al., 2005*) in the case of the *ARNOLDY2, ATCOPIA62, ATCOPIA95, TA12, TAG1* families, which do not contain copies with intact extremities in the Col-0 genome. Next, we selected all reads with one end (≥20 nt) mapping to a TE extremity (by locating reads where the CIGAR string contained only one 'S' character with a value equal or greater to 20). These reads were recursively soft clipped by 1 nt and mapped to the TAIR10 reference genome using Bowtie2 (using the arguments –end-to-end –mp 13 –rdg 8,5 –rfg 8,5 –very-sensitive) until the soft-clipped read length reached 20 nt. Read clusters composed of five or more reads clipped from the same extremity and overlapping with read clusters composed of reads clipped from the other extremity were taken to indicate the presence of a bona fide TE insertion only if the size of the overlap was equal or less than two-fold larger than that reported for TSDs for the corresponding TE family. Putative non-reference TE insertions overlapping genomic regions with aberrant coverage (as defined above) or located within inner pericentromeres (*Clark et al., 2007*) or spanning the corresponding donor TE sequence were filtered out. Presence or absence of these putative non-reference TE insertions was verified in other accessions by relaxing the parameters used to detect them in the first place. Specifically, we asked for the presence of a minimum of two rather than ten soft-clipped reads spanning the corresponding TSD coordinates. This improved the discovery of putative TE-insertions that are shared by more than one accession.

## Estimation of sensitivity

The rate of false negatives was estimated indirectly by first searching for TE insertions with TSDs that are present in the Col-0 reference genome but absent from the Ler-1 assembled genome sequence recently obtained using PacBio long-reads (unpublished). Specifically, annotated TE sequences together with 1 kb of upstream and downstream sequences were aligned using BLAT to the Ler-1 genome sequence. Heavily truncated TE sequences (covering less than 50% of full-length copies) were not considered, as most of these are unlikely to have been mobilized recently. Out of the 7931 TE sequences analyzed, 200 are not present in the Ler-1 genome sequence and among these, 142 have TSDs (*Figure 1—source data 4*). This value was then compared to that obtained by mapping Col-0 Illumina short split-reads to the Ler-1 genome sequence. A total of 64 TE insertions with TSDs could not be detected using that approach, thus giving a false negative rate of 45% (*Figure 1—figure supplement 2C*). Furthermore, 15 TE insertions were called that are in fact not present in the Col-0 genome sequence, thus giving a false positive rate of 10%; (i.e. FDR of 15.3%; *Figure 1—figure supplement 2C*).

## Manual filtering of residual false positives

We analyzed two re-sequenced Col-0 genomes (*Schmitz et al., 2013*; *Long et al., 2013*) using our split-read pipeline. Non-reference TE insertions with TSDs detected in other accessions and also identified in at least one of the two Col-0 genomes were filtered out, as they are most likely errors. Additionally, after manual inspection using the Integrative Genomics Viewer (IGV; *Robinson et al., 2011*), we eliminated all putative non-reference TE insertions with TSDs for which read-clusters comprise a disproportionately high number of reads (more than three MADs over the median whole genome coverage) as well as all putative non-reference TE insertions with TSDs that are present in 50% of accessions, as these are frequently the result of sequencing errors or mapping artifacts. Indeed, the pattern of 5' and 3' split-reads at the insertion site differs radically between false and

*bona fide* non-reference TE insertions, which enable their discrimination by visual inspection (*Figure 1—figure supplement 1*). We also observed a high rate of non-reference TE insertions with TSDs within TE sequences, mostly in the case of the *ATHILA* and *GYPSY* families. Visual inspection of these insertions indicates that they most likely result from mapping artifacts and they were therefore excluded. Finally, we examined 12 non-reference TE insertions with TSDs by PCR and all were validated (*Supplementary file 3*).

## TE sequence capture

A complete description of our TE-capture design and protocol will be published elsewhere. Briefly 2.1M biotinylated capture probes (Roche-NimbleGen) were designed to cover 200bp at each end of 310 potentially mobile TEs (*Supplementary file 4*), belonging to 181 of the 326 TE families annotated in TAIR10. These 181 TE families include 117 of the 131 mobile TE families identified using the split-read approach as well as most other TE families for which non-degenerate and thus potentially mobile copies are present in the Col-0 genome. Mobile or potentially mobile TE families with skewed sequence composition or very high copy number (including all ATHILA families) were excluded from the design in order to avoid overt biases in the capture of TE sequences. Twelve DNA sequence libraries for the accessions Pi-0, Bl-1, Sp-0, Ta-0, Lip-0, Bla-1, Stw-0, Cvi-0, Kondara, Jm-0, Gre-0 and Rubezhnoe-1 were prepared following the Illumina TruSeq paired-end kit, with some modifications. One µg of genomic DNA was sheared to a median fragment size of 400-450 bp. Fragments were subjected to end repair, A-tailing and index adapter ligation following the manufacturer's instructions. Libraries were then amplified through 7 cycles of ligation-mediated PCR using the KAPA HiFi Hot Start Ready Mix and primers AATGATACGGCGACCACCGAGA and CAAGCAGAAGACGGCATACGAG at a final concentration of 2 µM. PCR products were cleaned-up using the Qiaquick PCR Purification kit following the manufacturer's instructions, except that DNA was eluted in PCR grade water. Amplified DNA libraries were then pooled and one µg of the multiplex pool was used in the first hybridization step to capture probes (72 hr). Captured DNA was recovered using Streptavidin Dynabeads, washed and PCR amplified (5 cycles) as above. Amplified captured DNA was then subjected to a second round of hybridization (12 hr), recovery and amplification (14 cycles). Hybridization, captured DNA recovery and washing were performed as described in the NimbleGen SeqCapture EZ protocol. Pre-capture and post-second capture PCR products were run on an Agilent Bioanalyser DNA 1000 chip. Enrichment for capture TE sequences was confirmed by qPCR and estimated at 50–100 fold depending on the TE. Pair-end sequencing was performed with Illumina HiSeq 2000 and 100 bp reads. Between 10.7 and 16.1 million pairs were sequenced per library. Pairs were mapped to the TAIR10 reference genome using Bowtie2 with the arguments –mp 13 –rdg 8,5 –rfg 8,5 –very-sensitive -X 1000. . Discordant pairs were remapped using Bowtie2 with the parameters –mp 13 –rdg 8,5 –rfg 8,5 -D 20 -R 10 -N 0 -L 15 -i S,1,0.50 -k 100. TE insertions were detected in each accession using discordantly mapped reads and the algorithm Hydra (*Quinlan et al., 2010*) using the arguments: pairDiscordants: -d 50000 –z 1000 –x0 –r 100; dedupDiscordants: -s3; hydra: -ms 50 –li –use all –mld (10*mad) –mno (median+(20*mad). Enrichment factor was ~250 fold on average (*Figure 1—figure supplement 2E*). Comparing the results of TE-capture with those obtained using our split-read pipeline gives respectively false negative and positive rates of 56% and 7% (i.e. FDR of 14%; *Figure 1—figure supplement 2E*). These values are similar to those obtained using the assembled genome sequence of Ler-1 and confirm the low sensitivity of the split-read pipeline.

## GWAS of CNVs

Unlike for the initial CNV analysis, we considered this time only the annotated TE sequences that are at least half the size of the corresponding reference TE sequence (obtained from the *A. thaliana* RepeatMasker repeat library, http://repeatmasker.org) as this improved significantly the correlation between CN and the number of non-reference TE insertions. For each of the 113 families identified as mobile by both the split-reads approach and TE-capture, aggregated CN was obtained for each accession by summing the CNs estimated for each annotated copy. GWAS on these CN values was carried out with imputed SNPs (MAF > 5%) from the 211 accessions collected worldwide (*Schneeberger et al., 2011*; *Schmitz et al., 2013*), the 180 Swedish accessions (*Long et al., 2013*) and the joined dataset using a linear mixed model (LMM) (*Kang et al., 2010*). Kinship matrix was

included in the model as a random effect to control population structure. Out of the 113 TE families, 33 displayed a genome inflation factor (GIF) greater than 1.10 and were excluded from subsequent analysis. A conservative threshold value (p-value < $1 \times 10^{-8}$) was set to call statistically associated SNPs. Proximal associated SNPs in linkage disequilibrium ($r^2 > 0.2$) were identified using Plink (*Purcell et al., 2007*) and combined in blocks to build statistical associated intervals, which were expanded by 1 or 5 kb on either side. Variance explained by the leading SNP within each locus was calculated using the following equation: $\sigma^2_{SNP} = 2(MAF)(1 - MAF)\frac{\beta^2}{\sigma^2_y}$, in which MAF is the minor allele frequency, β is the SNP effect estimated by the LMM and $\sigma_y^2$ is the variance of the phenotype Y. Total variance explained by *cis* and *trans* loci was computed as the sum of the single-locus explained variances, under the assumption of additive contributions. The probability P(f) of missing a non-reference TE insertion with TSD underlying a 'false' *trans* locus through both the split reads and TE-capture approaches was calculated using the following equation:

$$P(f) = FN \times \binom{n}{12-k} \times f^{12-k} \times (1-f)^k$$

where f is the frequency of the minor allele for the *trans* locus under consideration (reported in the *Supplementary file 1*), FN is the false negative rate for the split-reads bioinformatic pipeline (0.56), n is the number of accessions analyzed by TE-capture (12) and k is the number of accessions without the non-reference TE insertion with TSD among the 12 accessions analyzed by TE-capture (k ranges from 0 to 12 and calculations were all performed using k=12).

## Climate analysis

We selected 12 geo-climatic variables representing different ecological layers: Aridity, number of frosty days, number of consecutive frost-free days, day length in the spring, maximum temperature in the warmest month, minimum temperature in the coldest month, temperature annual range, photosynthetically active radiation, precipitation in the wettest month, precipitation in the driest month, relative humidity in the summer, landscape slope and thermal (*Hancock et al., 2011*). CNs for the 113 families confirmed as mobile by the split-read approach and TE-capture were used to calculate a partial Mantel correlation with the 13 geo-climatic variables. Population structure kinship was included in the test to control population stratification. Partial Mantel tests were conducted using the 'ecodist' package in R. A threshold of p<0.01, corrected for multiple testing ($8.33 \times 10^{-4}$), was set to call statistically correlated variables. In addition to using the partial Mantel test, we also applied linear models to regress CN as a function of the climatic variables. Although this method does not control for population structure, it largely confirmed the associations found by the partial Mantel test.

## Characterization of *cis* and *trans* associations

Protein coding genes, miRNA genes, ncRNAs and TE annotations were retrieved from (ftp://ftp.arabidopsis.org/home/tair/Genes/TAIR10_genome_release/TAIR10_gff3/TAIR10_GFF3_genes.gff). GWAS intervals were defined as *cis* associations if they overlap with a TE annotation or a non-reference TE insertion with TSDs of the same family. All other GWAS intervals were defined as *trans* associations. The *cis* associations are over-represented 34 times (p-value $<1 \times 10^{-16}$) when compared to randomly chosen genomic intervals of the same size, which is consistent with TE activity being primarily determined by the TE sequence itself. All genomic annotations overlapping with *trans* intervals (within 1 or 5 kb) were considered as being putatively causal.

## Analysis of the localization of non-reference TE insertions with TSDs along chromosomes

To assess if non-reference TE insertions with TSDs are enriched in pericentromeric regions, their number within these regions was compared with that expected from a random distribution. Insertions with different allele counts (private, shared by 2–10 accessions, shared by >10 accessions) were considered separately. TE distribution in the reference genome (TAIR10) was obtained by counting the number of TE sequences located within pericentromeres (minus genomic regions showing aberrant coverage and inner pericentromeres). The expected distribution for the 2835 non-reference TE insertions with TSDs was calculated by randomizing $10^6$ times their position across the chromosomes

(genomic regions showing coverage deviation, the inner pericentromeres, or coordinates spanning the corresponding reference TE sequence were excluded). This set of random positions was used as a control for all subsequent analyses. Insertion distribution over genes and neighboring sequences was performed using a meta-gene. Briefly, protein coding gene features were extracted from the TAIR10 annotation and coordinates of non-reference TE insertions with TSDs were crossed with the set of genic features according to the following stepwise hierarchy: 5' UTR > 3' UTR> exon > intron > intergenic regions. For insertions that do not overlap protein-coding genes, the distance to the closest gene was calculated and reported as negative or positive distance according to the gene orientation. Expected insertion distribution under the null hypothesis was retrieved by applying this procedure for each of the $10^6$ randomized sets of insertions. To assess if non-reference TE insertions with TSDs are enriched within small clusters, we divided the genome into 10 kb non-overlapping windows and we counted the number of insertions events within them. The observed and expected (random) densities of non-reference TE insertions with TSDs per window were compared and significant enrichment was declared when the number of insertions found within a window was in the upper 0.005% tail of the random distribution.

## Reconstruction of the historical recombination landscape

Historical recombination was estimated using LDhat (*McVean et al., 2004*) as described before (*Choi et al., 2013*). Briefly, biallelic SNPs (MAF $\geq$ 0.1) from the 210 accessions collected worldwide were selected and split into blocks of 5000 SNPs with overlap of 500 SNPs. SNPs located in inner pericentromeres were excluded from the analysis. Blocks of SNPs were formatted using the 'convert' program. A likelihood lookup table was generated for 210 individuals with program 'complete' using the following parameters: –n 210 –rhomax 100 –n_pts 100 –theta 0.001. Population-scaled recombination rate (ρ/kb, ρ = 4Ner, where Ne is the effective population size and r is the per-generation recombination rate) was calculated using the 'interval' program with the following parameters: -its 60000000 –bpen 5 –sam 40000. Recombination rates for contiguous blocks were joined at overlap position 250. Population-scaled recombination rate map is provided in *Supplementary file 5*.

## Identification of DNA sequence motifs overrepresented at non-reference TE insertion sites with TSDs

Sequence spanning non-reference insertion sites were analyzed using Bioprospector Release 2 (*Liu et al., 2001*) using the following parameters: -r 1 -n 200 -a 1 -W 'TSD size'. Background sequence distribution for the reference genome was obtained using the 'genomebg' program. Sequence logo was produced using the seqLogo package version 1.36.0.

## Assessing the impact of non-reference TE insertions with TSDs on gene expression

All genes with a non-reference TE insertion with TSD within 1 kb were retrieved and their expression analyzed using transcriptome data available for 144 accessions (*Schmitz et al., 2013*). For each gene, we calculated the ratio between the median gene expression level for the accessions harboring the TE insertion and the median gene expression level for accessions lacking that insertion. Distribution plots of observed gene expression ratios were compared to the expected distribution under the null hypothesis (random effect). This expected distribution was obtained by calculating the gene expression ratio for $10^6$ randomly chosen sets of genes for which the TE insertion presence/absence 'label' was randomly assigned between the accessions. Statistical significance of differences between the observed and expected distributions was determined using the Kolmogorov-Smirnov test.

## Expression levels of genes affected by non-reference insertions with TSDs

RNA was extracted using the RNeasy plant mini kit (Qiagen) from plants grown under normal conditions (10 days old seedlings grown in liquid medium) or subjected to heat shock treatment (*Ito et al., 2011*). RT-qPCR was performed as described previously (*Silveira et al., 2013*). Primers details are given in *Supplementary file 3*. RT-qPCR results (one biological replicate only) are

indicated relative to those obtained for a gene (*AT5G13440*) that shows invariant expression under multiple conditions.

## Gene expression profile of genes affected by long-distance DNA methylation

Expression data in the Col-0 accession were obtained from http://www.weigelworld.org/resources/microarray/AtGenExpress/AtGE_dev_gcRMA.txt.zip/at_download/file for the 224 and 162 genes affected respectively by short- and long-distance DNA methylation in accessions with non-reference TE insertions with TSDs (by definition, these TE insertions are absent in Col-0). Triplicate data for each developmental time point was averaged and then normalized across the developmental time-point series. Average expression level was then calculated for each time point for all genes affected by short-distance DNA methylation and compared to the average calculated for all genes affected by long-distance DNA methylation. Statistical significance of differences between these two averages was calculated using the non-parametric Mann-Whitney U test.

## Determination of *FLC* haplotypes

Haplotype analysis was performed as described previously (*Li et al., 2014*). Briefly, SNPs within 100 kb of *FLC* were retrieved for the 211 worldwide accessions and used as input into fastPHASE version 1.4.0 (*Scheet and Stephens, 2006*). Default parameters were kept, except for the -Pzp option. For each SNP, haplotype membership with the highest likelihood was assigned.

## Code availability

Source code for the split-read pipeline can be accessed at https://github.com/LeanQ/SPLITREADER

# Acknowledgements

We thank members of the Colot lab and especially Mathilde Etcheverry for discussions. We thank Edith Heard and Pierre Capy for critical reading of an earlier version of the manuscript. This work was supported by the European Union Seventh Framework Programme Network of Excellence Epi-GeneSys (Award 257082, to VC), the Investissements d'Avenir ANR-10-LABX-54 MEMO LIFE, ANR-11-IDEX-0001-02 PSL* Research University and ANR-12-ADAP-0020-01 (to VC) and the Chaire Blaise Pascal (to RAM). LQ was the recipient of postdoctoral fellowships from the ANR-10-LABX-54 MEMO LIFE and ANR-11-IDEX-0001-02 PSL* Research University. ABS was the recipient of postdoctoral fellowships from the ANR-12-ADAP-0020-01 and from the Brazilian National Council for Scientific and Technological Development (CNPq). LQ and VC conceived and designed the study. LQ performed all of the bioinformatic analyses. ABS, GFM and JAJ designed the TE-sequence capture, ABS performed the experiment and CL and RAM performed the sequencing of captured fragments. ABS performed all of the other experimental analyses. LQ and VC wrote the paper, with additional input from all authors. Correspondence and requests for materials should be addressed to colot@biologie.ens.fr.

# Additional information

### Competing interests

GFM: declares a competing interest as employee of Roche NimbleGen Inc. JAJ: JAJ declares a competing interest as employee of Roche NimbleGen Inc. The other authors declare that no competing interests exist.

### Funding

| Funder | Grant reference number | Author |
| --- | --- | --- |
| Agence Nationale de la Recherche | ANR-10-LABX-54 MEMO LIFE | Vincent Colot |
| Agence Nationale de la Recherche | ANR-11-IDEX-0001-02 PSL* Research University | Vincent Colot |

| Agence Nationale de la Recherche | ANR-12-ADAP-0020-01 | Vincent Colot |
| European Commission | EpiGeneSys (Award 257082) | Vincent Colot |
| Conselho Nacional de Desenvolvimento Científico e Tecnológico | | Amanda Bortolini Silveira |
| Chaire Blaise Pascal | | Robert A Martienssen |

The funders had no role in study design, data collection and interpretation, or the decision to submit the work for publication.

## Author contributions

LQ, Conception and design, Acquisition of data, Analysis and interpretation of data, Drafting or revising the article; ABS, Acquisition of data, Analysis and interpretation of data, Drafting or revising the article; GFM, JAJ, Drafting or revising the article, Contributed unpublished essential data or reagents; CL, RAM, Acquisition of data, Drafting or revising the article; VC, Conception and design, Analysis and interpretation of data, Drafting or revising the article

## Author ORCIDs

Leandro Quadrana, http://orcid.org/0000-0001-6279-211X
Vincent Colot, http://orcid.org/0000-0002-6382-1610

# Additional files

## Supplementary files

• Supplementary file 1. Summary of GWAS results for CNV. Distribution of CN values, Manhattan plot and QQ-plot across the joined data set (391 accessions) for the indicated TE families. Summary statistics of associations are indicated below. MAF indicates Minor Allele Frequency in the joined dataset. Genes within GWAS intervals are indicated (*MET2a* is in bold).

• Supplementary file 2. DNA sequence motifs at insertions sites. Sequence logo of the overrepresented DNA sequence motifs at insertion sites ( ± 30bp) is shown for the 79 mobile TE families with at least 10 non-reference TE insertions with TSDs. The number of sequences used in each case is indicated.

• Supplementary file 3. PCR validation of non-reference TE insertions with TSDs and list of primer sequences used in this study.

• Supplementary file 4. List of TE-capture targets.

• Supplementary file 5. Historical population-scaled recombination rate map for *A. thaliana*

## Major datasets

The following dataset was generated:

| Author(s) | Year | Dataset title | Dataset URL | Database, license, and accessibility information |
| --- | --- | --- | --- | --- |
| Leandro Quadrana, Amanda Bortolini Silveira, George F Mayhew, Chantal LeBlanc, Robert A Martienssen, Jeffrey A Jeddeloh, Vincent Colot | 2016 | TE-capture sequencing data | http://www.ebi.ac.uk/ena/data/search?query=PRJEB11706 | Publicly available at the EBI European Nucleotide Archive (accession no: PRJEB11706). |

The following previously published datasets were used:

| Author(s) | Year | Dataset title | Dataset URL | Database, license, and accessibility information |
|---|---|---|---|---|
| Schmitz RJ, Schultz MD, Urich MA, Nery JR, Pelizzola M, Libiger O, Alix A, McCosh RB, Chen H, Schork NJ, Ecker JR | 2013 | Patterns of Population Epigenomic Diversity | http://www.ncbi.nlm.nih.gov/sra/?term=SRA012474 | Publicly available at the NCBI Sequence Read Archive (accession no. SRA012474) |
| Long Q, Rabanal FA, Meng D, Huber CD, Farlow A, Platzer A, Zhang Q, Vilhjálmsson BJ, Korte A, Nizhynska V, Voronin V, Korte P, Sedman L, Mandáková T, Lysak MA, Seren Ü, Hellmann I, Nordborg M | 2013 | Massive Genomic Variation and Strong Selection in Arabidopsis Thaliana Lines from Sweden | http://www.ncbi.nlm.nih.gov/sra/?term=SRA052536 | Publicly available at the NCBI Sequence Read Archive (accession no. SRA052536) |
| Schneeberger K, Ossowski S, Ott F, Klein JD, Wang X, Lanz C, Smith LM, Cao J, Fitz J, Warthmann N, Henz SR, Huson DH, Weigel D | 2011 | Reference-guided assembly of four diverse Arabidopsis thaliana genomes | http://1001genomes.org/data/MPI/MPISchneeberger2011/releases/current/Ler-1/Reads/ | Publicly available from the 1001 Genomes Data Center (http://1001genomes.org) |
| Schmitz RJ, Schultz MD, Urich MA, Nery JR, Pelizzola M, Libiger O, Alix A, McCosh RB, Chen H, Schork NJ, Ecker JR | 2013 | Patterns of Population Epigenomic Diversity | http://www.ncbi.nlm.nih.gov/geo/query/acc.cgi?acc=GSE43857 | Publicly available at NCBI Gene Expression Omnibus (accession no: GSE43857) |
| Schmitz RJ, Schultz MD, Urich MA, Nery JR, Pelizzola M, Libiger O, Alix A, McCosh RB, Chen H, Schork NJ, Ecker JR | 2013 | Patterns of Population Epigenomic Diversity | http://www.ncbi.nlm.nih.gov/geo/query/acc.cgi?acc=GSE43858 | Publicly available at NCBI Gene Expression Omnibus (accession no: GSE43858) |
| Stroud H, Greenberg MVC, Feng S, Bernatavichute YV, Jacobsen SE | 2013 | Comprehensive Analysis of Silencing Mutants Reveals Complex Regulation of the Arabidopsis Methylome | http://www.ncbi.nlm.nih.gov/geo/query/acc.cgi?acc=GSE39901 | Publicly available at NCBI Gene Expression Omnibus (accession no: GSE39901) |
| Ecker J | 2011 | Arabidopsis thaliana flowering time | https://www.arabidopsis.org/servlets/TairObject?type=germplasm&id=6530472136 | Available at Arabidopsis website (stock no: CS76636) |

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
