## [Decision Letter]

Thank you for submitting your article "The *Arabidopsis thaliana*mobilome and its impact at the species level" for consideration by *eLife*. Your article has been reviewed by three peer reviewers, and the evaluation has been overseen by a Reviewing Editor and Detlef Weigel as the Senior Editor. The following individuals involved in review of your submission have agreed to reveal their identity: Magnus Nordborg (Reviewer #3).

The reviewers have discussed the reviews with one another and the Reviewing Editor has drafted this decision to help you prepare a revised submission.

Summary:

This is a very interesting study that demonstrates extensive activity of many types of transposable elements (TEs) in the *A. thaliana* genome. It shows that TE copy number can be treated as a quantitative trait and that loci influencing it can be mapped. It also shows tantalizing evidence that TE copy numbers for some families are correlated with climate, which may provide insight into function and evolution. It provides strong evidence that TEs generally insert randomly across the genome, and that the strong bias towards pericentromeric regions is likely a consequence of purifying selection. This paper also provides evidence that many of the DNA methylation polymorphisms identified in *A. thaliana* are caused by TE insertions, which show complex interactions with nearby genes.

Essential revisions:

The reviewers raised a number of concerns that must be adequately addressed before the paper can be accepted. Some of the revisions will require limited additional analyses.

1) The manuscript is difficult to follow in many places. Some of the figure panels (1B-D, 3D-E, 4E) are too small, poorly explained, or both. For each TE family in Figure 1 there should be at least one accession in which the CNV is log 0, but this is not apparent at the presented resolution. The absolute numbers of predicted TE insertions should also be provided. Figure 1 is not clear and a short description without reference to Materials and methods would be helpful. The symbols in Figure 1 are not clear, and it would be helpful to have guides as to how the data relate to the narrative in the text – which data points make up the 1408 and 743 insertions? In Figure 3, it's not clear how the expected ratio is calculated. Figure 4 is too small to discern the depicted methylation patterns.

Furthermore, important points are sometimes covered in the text with little explanation. For example, the significance of the correlations between TE insertions, gene density and recombination rates is not explained, nor is the Hill-Robertson effect. The paper gives the overall impression of too much data presented in too little space. Please revise the manuscript to adequately explain all observations and analyses for a broad readership.

2) The conclusion that TEs insert with little bias may not be equally true for all TEs because insertion preferences are caused by TE-encoded transposase proteins. If your data allows such analyses to be performed with sufficient statistical power, please examine insertion preferences of TE families, or of Class I vs. Class II TEs.

3) The correlations between TE insertion and gene expression are potentially confusing because only two families of TEs are presented – one shows significant positive and negative associations, and one does not. Do these TE families represent the most extreme examples? An analysis of other TE families would be very helpful.

4) The presented data analysis is affected by two major biases: a false negative rate of nearly 50% and the fact that only insertions that are not in the Col-0 reference genome are considered. A false negative rate of over 45% can dramatically affect results of pair-wise comparisons of over 200 accessions. For example, when discussing the GWAS trans associations, the 45% false-negative rate may suggest that many of the trans associations are actually in cis. With this in mind, please discuss the limitations of you analytical approach. Furthermore, the estimated false positive rate (10%) from the PacBio comparison should be explicitly mentioned.

5) It is surprising that the cis and trans GWAS associations on average explain so much less variation than the numbers in Figure 2. Please discuss this point.

6) Confirming the correlation between TE insertions and methylation is very important, but discussion of this is surprisingly brief. When you write that "[f]or 55% of these sites, presence or absence of DNA methylation bears no relation to the presence or absence of the TE insertion. The remaining 45% of sites are methylated only when they contain the TE insertion, suggesting in this case a causal link", it sound dichotomous. Is correlation 0 for 55%, 1 for the rest? Given how noisy your data are (false positives and negatives), this cannot be true. So what are the numbers? And do you sometimes see strong methylation when there really is no trace of a TE?

7) Please discuss your conclusions in relation to those made in the manuscript by Stuart and colleagues "Population scale mapping of transposable element diversity reveals links to gene regulation and epigenomic variation" placed recently on bioRxiv (http://biorxiv.org/content/early/2016/02/21/039511).

---

## [Author Response]

1) The manuscript is difficult to follow in many places. Some of the figure panels (1B-D, 3D-E, 4E) are too small, poorly explained, or both. For each TE family in Figure 1 there should be at least one accession in which the CNV is log 0, but this is not apparent at the presented resolution. The absolute numbers of predicted TE insertions should also be provided. Figure 1 is not clear and a short description without reference to Materials and methods would be helpful. The symbols in Figure 1 are not clear, and it would be helpful to have guides as to how the data relate to the narrative in the text – which data points make up the 1408 and 743 insertions? In Figure 3, it's not clear how the expected ratio is calculated. Figure 4 is too small to discern the depicted methylation patterns.

*Furthermore, important points are sometimes covered in the text with little explanation. For example, the significance of the correlations between TE insertions, gene density and recombination rates is not explained, nor is the Hill-Robertson effect. The paper gives the overall impression of too much data presented in too little space. Please revise the manuscript to adequately explain all observations and analyses for a broad readership.*

We appreciate the reviewers’ comments and we have now clarified, extended and simplified many figures and conclusions.

Figure 1: We have added as a source file for all TE families the number of TE insertions expected based on our copy number estimates ([Supplementary-material SD1-data]). There is indeed for each TE family in Figure 1 at least one accession in which the CNV is log 0, and although this may not be apparent at the presented resolution (very pale yellow), the information is readily accessible in the source file.

Figure 1: We have enlarged this panel (now Figure 1) and added a schematic representation of the split-read pipeline (new Figure 1) that should facilitate the understanding of the rational of our approach. We have also expanded the figure legend of new Figure 1 to clarify what is shown. Finally, we have included an additional source file [Supplementary-material SD3-data]) containing the number of TE insertions for each TE family and superfamily to complement Figure 1 (now Figure 1).

Figure 3: we enlarged the two panels and have included in the main text the method used to calculate the expected distribution of gene expression ratios.

Figure 4: the panel is now enlarged (and now labeled Figure 4).

More generally, we have revised the manuscript to the best of our ability to explain all observations and analyses for a broad readership. Thus, we have expanded the Introduction slightly and we have broken the Results section into six sub-sections instead of four. Furthermore, our estimation of the sensitivity of the split-reads approach is now described in more detail and so are the results of the GWASs. We have also considerably simplified the explanation of the role of natural selection versus homologous recombination in shaping the TE insertion landscape in *Arabidopsis.* The Hill-Robertson interference is not mentioned anymore and we refer in the Discussion section to the review by Barton et al., 2014 for readers who would like to learn more about the potential roles of homologous recombination in the elimination of deleterious TE insertions.

*2) The conclusion that TEs insert with little bias may not be equally true for all TEs because insertion preferences are caused by TE-encoded transposase proteins. If your data allows such analyses to be performed with sufficient statistical power, please examine insertion preferences of TE families, or of Class I vs. Class II TEs.*

We have performed these analyses at the superfamily level, which provides sufficient statistical power in most cases. Results are presented in the text and in Figure 3—figure supplement 2.

3) The correlations between TE insertion and gene expression are potentially confusing because only two families of TEs are presented – one shows significant positive and negative associations, and one does not. Do these TE families represent the most extreme examples? An analysis of other TE families would be very helpful.

We now show the results for all of the TE superfamilies (Figure 3—figure supplement 3 and subsection “Impact of newly inserted TEs on the expression of adjacent genes”), although statistical power is only sufficient for the two superfamilies that were presented in the previous version of the manuscript.

*4) The presented data analysis is affected by two major biases: a false negative rate of nearly 50% and the fact that only insertions that are not in the Col-0 reference genome are considered. A false negative rate of over 45% can dramatically affect results of pair-wise comparisons of over 200 accessions. For example, when discussing the GWAS trans associations, the 45% false-negative rate may suggest that many of the trans associations are actually in cis. With this in mind, please discuss the limitations of you analytical approach. Furthermore, the estimated false positive rate (10%) from the PacBio comparison should be explicitly mentioned.*

We have taken these comments into consideration when presenting the composition of the mobilome (Results paragraph five), the distribution of non-reference TE insertions (subsection “**Genome localization of newly inserted TEs”**) and the results of the GWASs (paragraph five of subsection “**TE mobilization as a complex trait”**; Figure 2—figure supplement 3). However, as mentioned in the initial version of the manuscript and restated more clearly in the revised version (Results paragraph five), our estimate of the composition of the mobilome at the species level is not affected significantly by the low sensitivity of the split-reads approach thanks to the multiplicity of accessions used to for the analysis. Since our approach relies on a one-against-all comparison, we don’t feel that it is necessary to discuss pair-wise comparisons. Finally, the estimated false positive rate of the split-read is now explicitly indicated in the main text (Results paragraph four).

5) It is surprising that the cis and trans GWAS associations on average explain so much less variation than the numbers in Figure 2. Please discuss this point.

We have expanded the discussion of this surprising finding in the revised version.

*6) Confirming the correlation between TE insertions and methylation is very important, but discussion of this is surprisingly brief. When you write that "[f]or 55% of these sites, presence or absence of DNA methylation bears no relation to the presence or absence of the TE insertion. The remaining 45% of sites are methylated only when they contain the TE insertion, suggesting in this case a causal link", it sound dichotomous. Is correlation 0 for 55%, 1 for the rest? Given how noisy your data are (false positives and negatives), this cannot be true. So what are the numbers? And do you sometimes see strong methylation when there really is no trace of a TE?*

We have extended this part of the manuscript to provide a more detailed discussion of our findings (subsection “Impact of newly inserted TEs on the DNA methylation status of adjacent sequences”) and we have also included a new figure which shows the main patterns of DNA methylation observed at insertion sites (Figure 4). We have also added a figure that illustrates how the low sensitivity of the split-read approach can lead to discordant patterns of DNA methylation at a given insertion site among accessions containing the TE insertion (Figure 4—figure supplement 1). However, because the vast majority of non-reference TE insertions with TSDs are at very low frequency (i.e. are private or shared by two or three accessions only), correlation analyses between the presence or absence of a TE and the DNA methylation at the insertion site are not meaningful.

*7) Please discuss your conclusions in relation to those made in the manuscript by Stuart and colleagues "Population scale mapping of transposable element diversity reveals links to gene regulation and epigenomic variation" placed recently on bioRxiv (http://biorxiv.org/content/early/2016/02/21/039511).*

The work of Stuart and colleagues is cited twice in the new version of the manuscript to highlight important differences between their approach and ours and to complete the discussion about natural gene C-DMRs.